# Optimisation of AAV-NDI1 Significantly Enhances Its Therapeutic Value for Correcting Retinal Mitochondrial Dysfunction

**DOI:** 10.3390/pharmaceutics15020322

**Published:** 2023-01-18

**Authors:** Naomi Chadderton, Arpad Palfi, Daniel M. Maloney, Matthew Carrigan, Laura K. Finnegan, Killian S. Hanlon, Ciara Shortall, Mary O’Reilly, Peter Humphries, Lorraine Cassidy, Paul F. Kenna, Sophia Millington-Ward, G. Jane Farrar

**Affiliations:** 1Department of Genetics, The School of Genetics and Microbiology, Trinity College Dublin, D02 VF25 Dublin, Ireland; 2Department of Ophthalmology, Royal Victoria Eye and Ear Hospital, D02 XK51 Dublin, Ireland; 3The Research Foundation, Royal Victoria Eye and Ear Hospital, D02 XK51 Dublin, Ireland

**Keywords:** retinal ganglion cell (RGC), gene therapy, AAV, mitochondrial dysfunction, neuroprotection, oxidative stress, NDI1, complex I deficiency, LHON

## Abstract

AAV gene therapy for ocular disease has become a reality with the market authorisation of Luxturna^TM^ for RPE65-linked inherited retinal degenerations and many AAV gene therapies currently undergoing phase III clinical trials. Many ocular disorders have a mitochondrial involvement from primary mitochondrial disorders such as Leber hereditary optic neuropathy (LHON), predominantly due to mutations in genes encoding subunits of complex I, to Mendelian and multifactorial ocular conditions such as dominant optic atrophy, glaucoma and age-related macular degeneration. In this study, we have optimised the nuclear yeast gene, NADH-quinone oxidoreductase (NDI1), which encodes a single subunit complex I equivalent, creating a candidate gene therapy to improve mitochondrial function, independent of the genetic mutation driving disease. Optimisation of NDI1 (ophNdi1) substantially increased expression in vivo, protected RGCs and increased visual function, as assessed by optokinetic and photonegative response, in a rotenone-induced murine model. In addition, ophNdi1 increased cellular oxidative phosphorylation and ATP production and protected cells from rotenone insult to a significantly greater extent than wild type NDI1. Significantly, ophNdi1 treatment of complex I deficient patient-derived fibroblasts increased oxygen consumption and ATP production rates, demonstrating the potential of ophNdi1 as a candidate therapy for ocular disorders where mitochondrial deficits comprise an important feature.

## 1. Introduction

There is growing evidence that mitochondrial dysfunction, complex I deficiency and impaired oxidative phosphorylation (OXPHOS) may be involved in many neurodegenerative disorders including Alzheimer’s disease [1], Parkinson’s disease [2], Huntington’s disease [3], dominant optic atrophy (DOA) [4,5], Leber hereditary optic neuropathy (LHON) [6], glaucoma [7,8], diabetic retinopathy [9,10] and age-related macular degeneration (AMD) [11,12], among others. The retinal degenerations LHON, DOA and glaucoma primarily affect retinal ganglion cells (RGCs) and their axons, which form the optic nerve (ON), leading to RGC death, ON atrophy and sight loss in patients [8,13,14,15]. RGCs have high-energy demands, particularly in the unmyelinated prelaminar and laminar regions of RGC axons [16,17,18]. This high energy demand makes RGC axons highly vulnerable to mitochondrial dysfunction [19,20,21]. Compounding this, dysfunctional mitochondria produce high levels of reactive oxygen species (ROS), which inflict further oxidative damage on the cells.

LHON is a mitochondrially inherited eye disorder that causes significant visual disability in approximately 1 in 30,000–50,000 people [6,22,23]. RGC loss in LHON results from decreased ATP synthesis and increased ROS production, mostly caused by mutations within the seven mitochondrial-encoded subunits of complex I of the electron transport chain (ETC). Three of these mutations, 3460G>A (*ND1*), 11778G>A (*ND4*) and 14484T>C (*ND6*), account for approximately 90 percent of cases [24,25,26]. DOA affects 1 in 12,000–25,000 people [27,28] and is predominantly due to mutations in the nuclear-encoded *OPA1* gene [29,30]. This disrupts the finely tuned balance between mitochondrial fission and fusion resulting in deficits in cellular bioenergetics leading to RGC death [31,32]. Glaucoma is one of the leading causes of blindness worldwide [33] and is considered to have both genetic and environmental components [34]. It is characterised by damage to the optic nerve head, followed by apoptotic RGC death. Oxidative stress and mitochondrial dysfunction are thought to play key roles in glaucomatous RGC degeneration [35] and glaucoma patient lymphocytes have exhibited mitochondrial defects, including reduced complex I activity [36]. Importantly, in all conditions categorised by RGC loss a vicious cycle ensues, whereby RGC apoptosis itself generates ROS further impacting RGC health. 

In contrast to the 45-subunit mammalian complex I, which is encoded in both the nuclear and mitochondrial genome, the yeast complex I equivalent, NADH-quinone oxidoreductase (NDI1), is a single subunit nuclear-encoded gene. Ndi1 protein is transported to the mitochondria via an endogenous mitochondrial localisation signal (MLS). Therefore, an NDI1 gene-based therapy would not compete with endogenous mutant subunits for incorporation into the 45-subunit complex I, which may provide an advantage over therapies utilising single replacement subunits. This is particularly relevant where complex I dysfunction arises from mutations in a chaperone protein gene, or similar, which is integral to the complex I repair mechanism, e.g., DNAJC30 in recessive LHON [37]. NDI has previously been proposed as a potential mutation and gene-independent therapeutic in disease models of complex I deficiency, including models for Parkinson’s disease, LHON and demyelinating optic neuritis associated with multiple sclerosis [38,39,40,41], and may provide a potential method of treating secondary complex I deficiency conditions such as DOA and glaucoma.

We have previously demonstrated the potential of adeno-associated virus 2/2 (AAV)-delivered NDI1 to protect RGCs in a rotenone-induced murine model of optic neuropathy [39], which simulates many features of mitochondrial dysfunction. In this study, we used in silico analyses to optimise NDI1 codon usage to enhance expression in mammalian cells and codon substitution to mitigate potential immunogenicity, creating ophNdi1. Enhancing transgene expression by codon optimisation should allow for a significant reduction in the therapeutic dose required to achieve benefits, reducing the potential of an immune response when delivered virally [42]. We demonstrate that ophNdi1 functions significantly more efficiently than wild type NDI1 in decreasing ROS and increasing cellular bioenergetics in vitro, counteracting two of the hallmarks of mitochondrial dysfunction. When evaluated in the rotenone-induced mouse model, intravitreal injection of AAV-ophNdi1 significantly reduced RGC death and led to a significant preservation of retinal function as assessed by optokinetics (OKR) and photopic negative response (PhNR). Significantly, AAV-ophNdi1 also rescued the reduced ATP and oxygen consumption rates (OCR) observed in LHON patient-derived fibroblasts. The ophNdi1 treatment we propose here may also be more broadly applicable to other disorders with mitochondrial dysfunction.

## 2. Materials and Methods 

### 2.1. Codon Optimisation

A total of 329 codons of the NDI1 gene (*S. cerevisiae*) were modified to the most frequently used mammalian ones to optimise expression, while maintaining wild type amino acids. A Kozak sequence was also added to improve translational activity of the mRNA transcript. 

### 2.2. Predictions of Immunogenic Codons

In silico modelling of antigen presentation via the MHC-I pathway using the IEDB proteasomal cleavage/TAP transport/MHC class I combined predictor [43] was used to estimate immunogenicity. As fragments of 9 amino acids (9mer) in length are the most commonly presented fragments by MHC-I, all possible 9mers that could be derived from the NDI1 gene were analysed by the IEDB predictor. An immunogenicity value G*p*,*i* was generated for every 9mer peptide ‘*p*’ and MHC-I allele ‘*i*’. This took into account proteasomal degradation, transport and MHC-1 binding and was proportional to the amount predicted to be displayed on the cell surface. An overall immunogenicity factor F*p* for the 9mer peptide was then calculated using the following equation:$$F_{p}=\overset{i}{\sum }G_{p,i}N_{i}$$

*N_i_*: estimated prevalence of each allele in the global human population as a fraction of the total pool of alleles, calculated using population frequency data from The Allele Frequency Net Database [44]. An immunogenicity score SA was assigned to each amino acid position A in the peptide, defined as the sum of the immunogenicity factors for all 9mer peptides containing that amino acid. A BLOSUM matrix [45] was used to identify potential conservative mutations, predicted to be minimally disruptive to the structure and function of the Ndi1 protein, for all positions whose immunogenicity score was greater than one-fifth of the highest score. For all possible mutations at a given position, ΔB was defined as the change in the BLOSUM score for that mutation. All mutations for which ΔB was greater than 4 were considered too disruptive to protein function and not analysed further. For all remaining candidate mutations, immunogenicity factors F and scores S were recalculated for the post-mutation peptide using the IEDB predictor. The reduction in immunogenicity ΔS was then determined, defined as the difference between the score S for that position in the original peptide versus the new score S after mutation. All possible mutations were then ranked by the metric ΔS/ΔB, immunogenicity reduction/BLOSUM change ratio. 

### 2.3. Vector Construction

Codon optimised NDI1 sequences, designed to encode specific amino acid changes to reduce immunogenicity profiles and a minimal polyadenylation signal, were synthesised by Geneart, Inc. (Invitrogen, Paisley, UK) and cloned into pAAV-MCS (Agilent Technologies, CA, USA) downstream of a CMV promoter. The lead construct, pAAV-ophNdi1, contained 329 synonymous codon modifications and an I82V amino acid substitution (patent no. 10220102). An additional construct, pAAV-ophNdi1-HA, was created by cloning ophNdi1 with a C-terminal HA tag (synthesized by GeneArt®, Thermo Fischer Scientific, MA, USA) into pAAV-MCS. To enhance AAV packaging efficiency, 4.4 kb of bacteriophage lambda DNA [46] was inserted into the plasmid backbone of pAAV-ophNdi1 and the antibiotic resistance gene substituted for kanamycin. Cloning was verified by DNA sequencing. pAAV-Ndi1 and pAAV-CAG-EGFP vectors were cloned as previously described in [39,47], respectively. 

### 2.4. Cell Culture

Human embryonic kidney cells (HEK293, accession number CRL-1573; ATCC, Gaithersburg, MD, USA) were transfected with pAAV-ophNdi1, pAAV-Ndi1, pAAV-EGFP, pAAV-MCS or pAAV-ophNdi1-HA using Lipofectamine 2000 reagent (Thermo Fischer Scientific, MA, USA), according to the manufacturer’s instructions. Briefly, 5 × 10^5^ cells per well were seeded into 6-well plates and transfected 24 h later with 1 μg pAAV-Ndi1, the copy number equivalent of pAAV-ophNdi1, or 1 μg pAAV-MCS or 1μg pAAV-EGFP in Opti-MEM reduced serum media (Thermo Fischer Scientific, MA, USA). At 4 h post-transfection, media was replaced with complete DMEM. For the co-localisation study, 2 × 10^4^ cells per well were seeded onto 24-well plates containing a polylysine coated coverslip in 250µL complete DMEM and transfected with 1 μg pAAV-ophNdi1-HA. At 24 or 48 h post-transfection, cells were harvested or fixed in 4% paraformaldehyde. Mitotracker™ green (Thermo Fischer Scientific, MA, USA) was incubated with the cells for 30 min, according to the manufacturer’s instructions, when used. Each experiment was repeated in triplicate. 

### 2.5. Culture of Primary Human Dermal Fibroblasts 

Patient-derived fibroblasts were generated from biopsies taken from three male clinically and genetically diagnosed LHON patients carrying the G11778A ND1 LHON mutation (patient A, 36 years; patient B, 33 years and patient C, 40 years). Male age-matched control fibroblast cell lines were established from individuals with no history of mitochondrial or visual dysfunction (control 1, 26 years; control 2, 27 years and control 3, 35 years). 3 mm diameter skin punch biopsies were obtained from LHON patients and unaffected individuals who had given prior informed consent using a sterile biopsy punch (Inegra™ Miltex™, Integra Lifesciences Corp., Princeton, NJ, USA). Biopsies were cut into approximately 6 pieces and transferred to 4-well plates (Thermo Fisher Scientific, Waltham, MA, USA) containing DMEM supplemented with 20% FBS, 1% MEM non-essential amino acids, 1% sodium pyruvate, 1% penicillin-streptomycin and 1% amphotericin B (Sigma Aldrich). Plates were kept in a humidified incubator (37 °C, 5% CO_2_), with half of the media changed every 2–3 days. The media volume was maintained at a level that ensured biopsy pieces were in contact with the surface of the well, until cell outgrowth was observed. Upon reaching confluency (3–4 weeks), fibroblasts were subcultured using TrypLE Express (Thermo Fisher Scientific, MA, USA). Fibroblasts were subsequently maintained in DMEM supplemented with 10% FBS, 1% MEM NEAA and 1% sodium pyruvate and split (1:3) when 80% confluent. Cells were confirmed to be mycoplasma negative using the ‘LookOut^®^ mycoplasma PCR detection kit’ (Merck KGaA, Darmstadt, Germany), according to the manufacturer’s instructions. The harvesting of fibroblasts from human skin biopsies for the culture of fibroblasts was approved by the Royal Victoria Eye and Ear Hospital Research Ethics Committee (ref. no.: RF024/17).

### 2.6. NADH Oxidation Assay 

A total of 1.5 × 10^5^ HEK293 cells per well were seeded in triplicate in 6-well tissue culture dishes and transfected with pAAV-ophNdi1 or pAAV-Ndi1 24 h later, as described above. At 72 h post-transfection, mitochondria were isolated from cell samples via mechanical disruption (Dounce homogenizer) and NADH oxidation activities measured at 340 nm, as described in [48] (UV-mini 1280, Shimadzu, Kyoto, Japan). Additionally, 10 µL of flavone (50 µM) was added to inhibit Ndi1 or ophNdi1 protein activity to determine background levels of NADH oxidation. NADH oxidation activity was calculated (UVprobe software, Shimadzu, Kyoto, Japan) after each inhibitor addition to establish NADH oxidation activity specific to either complex I or Ndi1/ophNdi1. NADH oxidation activity was normalised using the Pierce™ detergent compatible Bradford assay kit (Thermo Fischer Scientific, MA, USA), with absorbance determined at 595nm (FLUOstar OPTIMA; BMG Labtech, Aylesbury, UK).

### 2.7. ROS Assay 

A total of 1.5 × 10^5^ HEK293 cells per well were seeded in a 24-well plate and transfected 24 h later with pAAV-MCS, pAAV-ophNdi1 or pAAV-Ndi1, as described above. At 24 h post-transfection, cells were treated with 0 nM or 5 nM rotenone. Another 24 h later, cells were harvested, washed and cell pellets suspended in 250 µL HBSS with 2.5 µL DNase (Merck KGaA, Darmstadt, Germany) and incubated with 2 µL CellROX™ Green Reagent (Invitrogen™, Thermo Fisher Scientific, MA, USA) or 2 µL MitoSOX™ Red Reagent (Invitrogen™, Thermo Fisher Scientific, MA, USA) in glucose free media without phenol red for 30 min at 37 °C. Following this, cells were strained through 0.50 µM filters (CellTrics; Sysmex Europe SE, Norderstedt, Germany) and 0.5µL of DRAQ5™ (BD Biosciences, NJ, USA) was added to aid in gating live cells (BD Accuri^TM^ C6; BD Biosciences, NJ, USA). Median levels of fluorescence in live cells (CellROX^TM^/MitoSox^TM^ positive, DRAQ5™ positive), representing relative ROS levels, were recorded. Transfected cells that had not been treated with rotenone were used to calculate the percentage change in ROS. 

### 2.8. Seahorse Mitochondrial Metabolism Assays 

A total of 5 × 10^3^ cells per well were seeded into Agilent XFe96 Seahorse plates, transduced 24 h later with pAAV-ophNdi1 or pAAV-Ndi1, at an MOI of 3.4 × 10^5^. A mitochondrial stress test was performed 48 h later according to the manufacturer’s protocols (Seahorse XFe96 extracellular flux analyser, Agilent Technologies, CA, USA). Injection cycles were 5× for basal OCR, 5× following oligomycin (1 µM), 5× following FCCP (1 µM), 5× following rotenone (0.5 µM) and 5× following antimycin A (0.5 µM) injections [4,49]. Six replicas per group were analysed. Basal and maximal OCRs, SRC and OCR rescue post-rotenone treatment (measurement 16–20/measurement 11–15 × 100) were determined. Primary fibroblasts were analysed as described above, but with 2.25 µM FCCP. The ATP Rate Assay was carried out as per the manufacturer’s protocol (Agilent Technologies, CA, USA) using 5 × 10^3^ cells per well. Seahorse analyses were normalised using the Pierce™ detergent compatible Bradford assay kit (Thermo Fischer Scientific, MA, USA), with absorbance determined at 595 nm (FLUOstar OPTIMA; BMG Labtech, Aylesbury, UK) and a standard test sample included on all Seahorse plates in a given experiment.

### 2.9. Adeno-Associated Virus (AAV)

AAV-ophNdi1 (patent no. 10220102), AAV-Ndi1 [39] and AAV-CAG-EGFP (AAV-EGFP; [47]) recombinant AAV2/2 viruses were generated as previously described [50]. Briefly, HEK293 cells were transfected with pAAV-ophNdi1, pAAV-Ndi1 or pAAV-EGFP, pRep2/Cap2 and pHelper (Agilent Technologies, Santa Clara, CA, USA) at a ratio of 1:1:2, using polyethylenimine. At 72 h post-transfection, AAV particles were purified from the clarified lysate by triple caesium gradient ultracentrifugation and dialysed against PBS supplemented with Pluronic F68 (0.001%; [51]). Genomic titres (vg/mL) were determined by quantitative real-time PCR [52]. 

### 2.10. Intravitreal Injection 

All animal work was performed in accordance with the European Union (Protection of Animals used for Scientific Purposes) Regulations 2012 (S.I. no. 543 of 2012) and the Association for Research in Vision and Ophthalmology (ARVO) statement for the use of animals and approved by the animal research ethics committee in Trinity College Dublin (Ref. no. 140514/240320). Animals were maintained in a specific pathogen free (SPF) facility. Adult wild type 129 S2/SvHsd mice (Harlan UK Ltd., Oxfordshire, UK) were anaesthetised by intraperitoneal injection of ketamine and medetomidine (57mg/kg and 0.5mg body weight, respectively). Pupils were dilated with 1% tropicomide and 2.5% phenylephrine, and topical anaesthesia (Amethocaine) administered. A 26-gauge microneedle attached to a 10 μL Hamilton syringe was inserted through a small puncture made in the sclera, and 3 µL AAV2/2 or 0.6 µL of 1.5 mM rotenone in DMSO was slowly administered into the vitreous over a two-minute period. An anaesthetic reversing agent (Atipamezole Hydrochloride, 1.33 mg/kg body weight) was delivered by intraperitoneal injection following injection. Body temperature was maintained during recovery. Animals were sacrificed by CO_2_ asphyxiation. 

### 2.11. RT-qPCR Analysis

Relative in vivo mRNA expression levels from AAV-ophNdi1 and AAV-Ndi1 injected murine retinas were determined by reverse transcription PCR (RT-qPCR). Adult mice (*n* = 6 eyes) were intravitreally injected with 3 × 10^9^ vg/eye AAV-ophNdi1 or 3 × 10^9^ vg/eye AAV-Ndi1. Four weeks post-injection, retinas were harvested, total RNA extracted and in vivo expression levels determined by real time RT-qPCR, as described in [53,54,55] (patent no. 10220102). The following primer pairs were used: ophNdi1 5’ GAACACCGTGACCATCAAGA 3′ and 5′ GCTGATCAGGTAGTCGTACT 3′, NDI1 5’ CACCAGTTGGGACAGTAGAC 3’ and 5’ CCTCATAGTAGGTAACGTTC 3’, β-actin 5′ AGAGCAAGAGAGGCATCC 3′ and 5′ TCATTGTAGAAGGTGTGGTGC 3′. Real time RT-qPCRs were performed twice in triplicate. Standard curves containing 1 × 10^2^–1 × 10^7^ copies/μL of ophNdi1 or NDI1 used to enable expression levels from all constructs, whether humanised or not, to be compared using absolute copy number. Expression levels were normalised using β-actin (5′ AGAGCAAGAGAGGCATCC 3′ and 5′ TCATTGTAGAAGGTGTGGTGC 3′).

### 2.12. Optokinetic (OKR) Analysis

Mice (*n* = 8 per group) underwent OKR analysis at 5 weeks post-rotenone injection as previously described [39]. OKR spatial frequency thresholds were measured blind using a virtual optokinetic system (VOS; OptoMotry, Cerebral Mechanics, AB, Canada; [56,57]). The spatial frequency threshold, the point at which the mouse no longer tracked, was obtained by incrementally increasing the spatial frequency of the grating at 100% contrast using the staircase procedure.

### 2.13. Photopic Negative Response 

Mice (*n* = 8–12 per group) underwent assessment of the cone ERG Photopic Negative Response (PhNR) at 7 weeks post-rotenone injection as previously described [39]. Mice were anaesthetised as described above. PhNR responses were recorded simultaneously from both eyes by means of goldwire electrodes (Roland Consulting, Brandenburg-Wiesbaden, Germany). Resulting waveforms were marked according to International Society for Clinical Electrophysiology of Vision conventions. Only those mice where a negative response was recorded in both eyes were included in the study.

### 2.14. Histological Analysis

Mice were sacrificed, eyes enucleated and fixed overnight in 4% paraformaldehyde in PBS before the retinas were removed from the eyecups and immunocytochemistry performed as described previously described [58]. Whole retinas were incubated with anti-BRN3A primary antibody (Synaptic Systems 411003, Goettingen, Germany; 1/200 dilution) [59,60] for 3 days at 4 °C, washed in PBS and then incubated with Cy3 conjugated secondary antibodies (Jackson ImmunoResearch Laboratories; 1/400) for 2 days at 4 °C. Wholemount and cell images were taken using an Olympus IX83 inverted motorised microscope (Mason Technology, Dublin, Ireland) equipped with a SpectraX LED light source (Lumencor, Mason Technology, Dublin, Ireland) and an Orca-Flash4.0 LT PLUS/sCMOS camera (Hamamatsu, Tsukuba City, Japan), as previously described [47]. Samples were imaged using a 4x (wholemount) and 40× (cells) objective utilising enhanced focal imaging (EFI) with typically 5–8 Z-slices. Lateral frames for wholemounts were stitched together and analysed in Olympus CellSens software (v1.9; Waltham, MA, USA). Cell counting was performed utilizing 2D deconvolution, manual threshold and object size filter, with the same settings/operations applied to all images. 

### 2.15. Statistical Analysis 

Statistical analysis included one way ANOVA with Tukey’s multiple comparisons test comparison post-hoc test and Student’s *t*-tests using Prism 9.3 (GraphPad), Datadesk 6.3.1 and Excel (MS); *p* < 0.05 was considered statistically significant.

## 3. Results

### 3.1. Optimisation of NDI1 

A total of 329 of the 513 NDI1 codons, representing the total number of codons requiring optimisation, were altered to those most frequently used in mammals to create a codon-optimised version of the gene (Appendix A). A Kozak sequence was also added to improve translation. Additionally, an in silico analysis was performed on all possible 9-mers that could be produced by proteasomal digestion of the peptide to identify sub-sequences most likely to be displayed by MHC-1 (relative predicted potential immunogenicity; Appendix A). Candidate amino acid substitutions were selected using a computational pipeline (Appendix A), with the goal of identifying changes which would have a large reduction in potential immunogenicity while having a relatively low impact on protein function (Appendix A). While not all regions contained amino acids that could be changed without significantly destabilising the Ndi1 protein, lead candidates were identified (Appendix A). I82V, the highest ranked candidate, was included in the codon-optimised version of the gene, creating ophNdi1 (Appendix A). 

### 3.2. ophNdi1 Functions More Efficiently Than Ndi1 In Vitro

Following in silico analysis, ophNdi1 was generated with a cytomegalovirus (CMV) immediate early promoter to enable direct comparison with wild type NDI1 [39]. OphNdi1 was initially evaluated in HEK293 cells as there are relatively few RGCs present in the murine retina (~60,000 RGCs in 129/J mice [61]), making it very difficult to harvest sufficient cells for analysis. 

#### 3.2.1. Mitochondrial Localisation and Expression of ophNdi1

The correct localisation of ophNdi1 to the mitochondria was confirmed in HEK293 cells by co-localisation of HA-tagged ophNdi1 and MitoTracker™ (Thermo Fisher), a fluorescent mitochondrial stain that localises to mitochondria independent of their membrane potential (Figure 1A–D). Notably, ophNdi1 includes an endogenous MLS. ophNdi1 and MitoTracker™ distribution patterns were shown to overlap completely (Figure 1C), suggesting that the endogenous MLS provided appropriate transport of ophNdi1 to the mitochondria. Importantly, normal tubular mitochondrial morphology is seen following expression of ophNdi1 in these cells (Figure 1D). Subsequently, the expression of ophNdiI and wild type NDI1 [39] were compared in vitro in HEK293 cells transfected with ophNdiI and NDI1 plasmids. The relative mRNA expression level of ophNdi1 (27,391.33 ± 3687.16; *n* = 6) was 19 times higher (*p* < 0.001, two sample *t*-test) than NDI1 (1444 ± 158.30; *n* = 6; Figure 1E).

#### 3.2.2. OCR 

Cellular oxygen consumption rates (OCR) provide an indirect measurement of mitochondrial activity, as oxidative phosphorylation (OXPHOS) is the primary method of oxygen consumption in mammalian cells. A decrease in OCR can indicate a cell under stress. AAV2/2 vectors were generated (AAV-ophNdi1 and AAV-NDI1), as this serotype efficiently transduces RGCs following intravitreal (IVT) delivery [39,62]. AAV was used for subsequent in vitro and in vivo delivery, and moreover, transduction tends to be less disruptive for the cell than plasmid transfection. OCRs of transduced and untransduced HEK293 cells were compared 48 h post-transduction (MOI = 1 × 10^5^). The basal OCR (Figure 1I) was significantly increased in cells transduced with AAV-ophNdi1 (237.97 ± 83.61 pmol/min) compared to untreated control cells (114.04 ± 35.66 pmol/min, *n* = 6; *p* < 0.05, ANOVA) and AAV-NDI1 transduced cells (109.53 ± 43.60 pmol/min, *n* = 6; *p* < 0.05, ANOVA). Furthermore, the maximal respiration (Figure 1J) was also significantly increased in AAV-ophNdi1 transduced cells compared to untreated control cells (control 126.71 ± 34.23 pmol/min and AAV-ophNdi1 242.89 ± 107.19 pmol/min; *p* < 0.05, ANOVA; *n* = 6). The spare respiratory capacity (SRC) was unchanged (ns), as both the basal and maximal rose by equivalent amounts leaving the difference between them, the SRC, unchanged. 

#### 3.2.3. ATP

Reduced ATP production is typically a consequence of mitochondrial dysfunction, therefore it was important to establish how ophNdi1 modulates this. HEK293 cells were transduced with AAV-ophNdi1 or AAV-NDI1 (MOI = 1 × 10^5^) and their ATP production levels were compared to untransduced control cells 48 h post-transduction (Figure 1K). AAV-ophNdi1 transduced cells demonstrated higher ATP levels (96.21 ± 35.79 pmol/min) compared to both the untransduced control cells (54.68 ± 19.75 pmol/min) and AAV-NDI1 transduced cells (58.76 ± 26.92 pmol/min) but did not quite reach significance (*n* = 0.0689 pmol/min, ANOVA; *n* = 6). ATP levels in AAV-NDI1 transduced cells were unchanged from control (ns), possibly due in part to insufficient expression in this model.

#### 3.2.4. Rotenone Resistance 

As the primary role of the optimised gene therapy is to substitute for complex I deficiency, we assessed the ability of ophNdi1 to protect cells from the effects of the irreversible complex I inhibitor, rotenone, by analysis of OCR. NDI1 had previously demonstrated significant rotenone insensitive respiration [39,63,64,65,66,67] hence it was important to determine the effect of the more highly expressed ophNdi1 in comparison to NDI1. HEK293 cells were transduced with AAV-ophNdi1 or AAV-NDI1 (MOI = 1 × 10^5^) and their OCRs analysed 48 h post-transduction. Following the addition of rotenone (0.5 µM), AAV-ophNdi1 transduced cells demonstrated significant rotenone-resistance (Figure 1L) compared to untransduced control cells (*p* < 0.0001, ANOVA). Following rotenone insult, AAV-ophNdi1-treated cells maintained 66.10 ± 11.86% of maximal OCR rates, while control cells only maintained 10.75 ± 3.46% (*p* < 0.0001, ANOVA). In contrast, AAV-NDI1 was not sufficiently highly expressed to rescue OCR post-rotenone insult in this model (OCR post-rotenone 8.56 ± 5.66%).

#### 3.2.5. NADH Oxidation and ROS

The ability of ophNdi1 and NDI1 to decrease ROS, a significant feature of mitochondrial dysfunction, and increase NADH oxidation activity, an indirect measure of complex I activity, was evaluated. To induce complex I deficiency and mitochondrial dysfunction, rotenone, a complex I inhibitor was utilised; an increase in ROS levels is typically a feature of rotenone-induced mitochondrial dysfunction [49,68,69]. Mitochondrial and cellular ROS levels were determined in ophNdi1 and NDI1 transfected HEK293 cells 24 h post-rotenone insult (5 nM) using MitoSOX™ and CellROX™ assays, respectively. ROS levels in transfected cells with and without rotenone insult were compared (Figure 1F,G). Notably, the rotenone-induced increase in mitochondrial ROS was significantly reduced in both ophNdi1 (*p* < 0.01, ANOVA) and NDI1 (*p* < 0.01, ANOVA) transfected cells compared to control cells (151.52 ± 15.01%, 153.25 ± 17.09% and 188.66 ± 25.51%, respectively; *n* = 8; MitoSOX™). OphNdi1-treated cells also exhibited a lower rotenone-induced increase in cellular ROS levels (113.90 ± 13.49% compared to 130.03 ± 10.00%, *n* = 8; *p* < 0.05, ANOVA; CellROX™). The data clearly demonstrate a reduction in both cellular and, particularly, mitochondrial ROS following rotenone insult when the cells expressed NDI1 or ophNdi1.

Complex I catalyses the oxidation of NADH to reduce ubiquinone, hence the measurement of NADH oxidation activity enables direct comparison of ophNdi1 and NDI1 in mitochondria isolated from HEK293 cells transfected with each construct (Figure 1H). Complex I activity was determined by measuring the rate of decrease in absorbance at 340 nm, which occurs when NADH is oxidised to NAD+, normalised to total protein [48]. The NADH oxidation activities in mitochondria extracted from ophNdi1 and NDI1 transfected HEK293 cells when endogenous complex I activity was inhibited with rotenone were 1489.44 ± 786.43% and 101.47 ± 37.95%, respectively, of control cells which had not been treated with rotenone (pAAV-EGFP transfected; 100 ± 11.88%; *n* = 3–5). Significantly, ophNdi1 NADH oxidation activity was up to 15-fold higher than endogenous NADH oxidation (*n* = 5; *p* < 0.01, ANOVA). Given that the levels of NADH oxidation mediated by NDI1 in the presence of rotenone were similar to endogenous levels in control cells that were not treated with rotenone, this result clearly demonstrates the ability of NDI1 to substitute for complex I activity when the latter is inhibited with rotenone. In contrast, control cells exhibited a significant reduction in NADH activity when administered rotenone (39.90 ± 4.57% of the untreated control, *n* = 4; *p* < 0.01, ANOVA). Notably, NDI1 acted synergistically with the endogenous complex I to give an increased NADH oxidation activity rate of 209.03 ± 97.36%, (*n* = 4; *p* < 0.01, ANOVA). 

### 3.3. ophNdi1 Provides Functional Benefit and Preserves RGCs In Vivo

Having demonstrated that ophNdi1 increases mitochondrial function, representing an equivalent for mammalian complex I with greater efficacy than NDI1 (Figure 1), it was important to evaluate whether AAV-ophNdi1 also protected against visual dysfunction and RGC loss in a complex I deficient (rotenone-induced) animal model. When rotenone is administered IVT to mice, it causes biochemical, structural and functional retinal deficits resembling those observed in patients, notably loss of RGCs and degeneration of the optic nerve [39,70,71,72]. 

#### 3.3.1. mRNA Expression In Vivo

Initially, expression of ophNdi1 was evaluated in vivo. Adult wild type 129 s2/SvHsd mice were injected IVT with 3 × 10^9^ vg of AAV-ophNdi1 or AAV-NDI1 and NDI1 expression levels compared in total retinal mRNA two weeks post-injection. Notably, the relative expression level of NDI1 mRNA from AAV-ophNdi1 (0.35 ± 0.16; *n* = 6) was ~30 times higher (*p* < 0.001, two sample *t*-test) than from AAV-NDI1 (0.01 ± 0.01; *n* = 6), confirming that the expression of ophNdi1 is indeed higher than NDI1, in vivo as well as in vitro. 

#### 3.3.2. Optokinetic Response

A further group of adult wild type mice received IVT injections in contralateral eyes of 3 × 10^9^ vg/eye AAV-ophNdi1 or PBS, and AAV-EGFP (1 × 10^8^ vg/eye, to facilitate identification of transduced regions of the retina). Six weeks post-injection, mice received 1.5 mM rotenone IVT in both eyes. At five weeks post-rotenone treatment, spatial vision was evaluated. Optokinetic responses (OKRs) were measured using a virtual optokinetic system [56]. The spatial frequency threshold, the point at which an animal no longer tracks the moving grating, was established separately for eyes treated with AAV-ophNdi1 and control eyes treated with PBS (Figure 2H). AAV-ophNdi1-treated eyes displayed significantly improved OKRs compared to control eyes post-rotenone insult (0.39 ± 0.05 cyc/deg and 0.16 ± 0.06 cyc/deg, respectively, *p* < 0.0001; ANOVA; *n* = 8–11). OKRs of un-insulted eyes that received AAV-ophNdi1 or wild type eyes were 0.43 ± 0.02 cyc/deg and 0.43 ± 0.03 cyc/deg, respectively. Notably, the OKRs of eyes treated with AAV-ophNdi1 and then insulted with rotenone maintained 91.25 ± 13.83% of wild type OKRs, while rotenone-insulted control eyes were reduced to only 36.62 ± 16.56% of wild type OKRs (*p* < 0.0001, ANOVA; *n* = 8–11). These data demonstrate that IVT administration of AAV-ophNdi1 preserved mouse vision in the presence of the complex I inhibitor rotenone. 

#### 3.3.3. Photopic Negative Response

To define the preservation of retinal function afforded by AAV-ophNdi1 following rotenone insult further, the photopic negative response (PhNR) was evaluated (Figure 2I,J) two weeks after OKR analysis (seven weeks post-rotenone treatment). The PhNR is the slow negative potential following the positive b-wave in photopic electroretinography that represents RGC function. AAV-ophNdi1-treated eyes displayed significantly better PhNR amplitudes compared to control eyes following administration of 1.5mM rotenone (7.15 ± 4.90 µV and 1.69 ± 1.75 µV, respectively, *p* < 0.05, ANOVA; *n* = 6–11). Eyes treated with AAV-ophNdi1 but no rotenone and wild type eyes had PhNRs of 8.51 ± 5.15 µV and 8.88 ± 3.29 µV, respectively. Hence, eyes that received AAV-ophNdi1 prior to rotenone administration did not exhibit significantly different PhNR amplitudes to wild type and AAV-ophNdi1-treated mice that had not received rotenone insult. These data indicate that AAV-ophNdi1 has indeed provided functional benefit in the presence of the complex I inhibitor rotenone. 

#### 3.3.4. RGC Preservation

RGC death is a hallmark of many ocular disorders and a feature of rotenone-induced complex I inhibition in the mouse model, therefore evaluation of RGC preservation represents a direct measure of the efficacy of AAV-ophNdi1. To evaluate whether the functional benefits mediated by AAV-ophNdi1 following rotenone insult (Figure 1) also resulted in RGC preservation, retinal wholemounts were prepared to enable RGC analysis throughout the whole retina (Figure 2A–G). Treated and control retinas were harvested eight weeks post-rotenone injection. BRN3A immunocytochemistry was used as an RGC marker and native EGFP, from the co-injected AAV-EGFP, was used to assess AAV transduction efficacy. Native EGFP expression was observed throughout the wholemount demonstrating that a significant proportion of the retina was transduced. Representative wholemounts from AAV-ophNdi1 alone, PBS (control) plus 1.5 mM rotenone and AAV-ophNdi1 plus 1.5 mM rotenone retinas are presented (Figure 2A–F). RGCs were counted to quantify the level of benefit obtained with AAV-ophNdi1 (Figure 2G). AAV-ophNdi1-treated eyes had a greater number of preserved RGCs following rotenone insult compared to control retinas (1783 ± 387.20 RGCs/mm^2^ and 1301.4 ± 28 RGCs/mm^2^, respectively, *p* < 0.01, ANOVA, *n* = 10–15). Greater numbers of RGCs were observed in retinas treated with AAV-ophNdi1 but not insulted with rotenone (2255.90 ± 17 RGCs/mm^2^, *p* < 0.05, ANOVA, *n* = 5) and in wild type un-injected retinas (3159.10 ± 15 RGCs/mm^2^, *p* < 0.0001, ANOVA, *n* = 5). These data demonstrate that intravitreal administration of AAV-ophNdi1 significantly preserved RGCs in the presence of the complex I inhibitor rotenone, though RGC numbers were approximately 30% lower than wild type un-injected retinas. 

### 3.4. AAV-ophNdi1 Provides Benefit in Complex I Deficient LHON Patient Cells 

Primary patient fibroblasts are a representative model to assess cellular bioenergetics [73,74]. Fibroblasts derived from three male patients (aged 33, 36 and 40 years) carrying the G11778A mutation in *ND4* known to cause complex I dysfunction and LHON were treated with AAV-ophNdi1 to determine if the beneficial effects seen in the rotenone-induced mouse model would lead to the phenotypic rescue in these patient-derived cells, representing a model of complex I deficiency. Patient fibroblasts were transduced with AAV-ophNdi1 at an MOI of 1 × 10^5^ and their basal OCR compared to untransduced patient cells and unaffected age-matched controls. Basal OCRs (Figure 3E; *n* = 5) were significantly increased in all three patient fibroblast lines (L1–L3) treated with AAV-ophNdi1; Patient L1 55.01 ± 18.50 pmol/min versus 40.76 ± 11.32 pmol/min (*p* < 0.05, paired *t*-test), Patient L2 67.37 ± 7.56 pmol/min versus 56.57 ± 3.46 pmol/min (*p* < 0.05, paired *t*-test), Patient L3 55.37 ± 12.20 pmol/min versus 43.50 ± 11.18 pmol/min (*p* < 0.05, paired *t*-test). The basal OCR of the unaffected control fibroblasts were 49.97 ± 6.68 pmol/min, 57.10 ± 6.36 pmol/min and 52.32 ± 5.24 pmol/min. Importantly, the maximal respiration (Figure 3F; *n* = 5) was significantly increased in two of the three patient fibroblast cell lines following treatment with AAV-ophNdi1: Patient L1: 74.97 ± 22.86 pmol/min versus 66.28 ± 15.02 pmol/min (*p* = 0.08, paired *t*-test), Patient L2: 72.25 ± 9.89 pmol/min versus 69.57 ± 8.92 pmol/min (*p* < 0.05, paired *t*-test), Patient L3: 74.55 ± 14.69 pmol/min versus 70.96 ± 13.79 pmol/min (*p* < 0.05, paired *t*-test). The maximal OCR of the unaffected control fibroblasts were 61.03 ± 11.70 pmol/min, 69.55 ± 12.46 pmol/min and 62.23 ± 12.34 pmol/min. The spare respiratory capacity (SRC), the difference between maximal and basal OCR, of patient fibroblasts was reduced following AAV-ophNdi1 treatment, although only significantly in the case of patient L2 (*p* < 0.05), as the basal OCR increase was somewhat greater than the increase in the maximal OCR leading to a reduction in the SRC. The SRC values were Patient L1: 26.43 ± 11.54 pmol/min post-treatment versus 32.90 ± 15.08 pmol/min pre-treatment, Patient L2: 9.61 ± 10.24 pmol/min post-treatment versus 18.80 ± 7.4 pmol/min pre-treatment and Patient L3: 29.44 ± 11.86 pmol/min post-treatment versus 32.52 ± 14.08 pmol/min pre-treatment. The SRC of the unaffected control fibroblasts were 13.24 ± 7.43 pmol/min, 14.15 ± 13.88 pmol/min and 15.15 ± 9.47 pmol/min. 

Notably, patient fibroblasts also showed higher ATP production levels following AAV-ophNdi1 treatment (Figure 3G; *n* = 5): Patient L1 42.54 ± 10.39 pmol/min versus 33.32 ± 10.30 pmol/min (*p* < 0.05, paired *t*-test), Patient L2 54.82 ± 3.85 pmol/min versus 46.21 ± 2.53 pmol/min (*p* = 0.06, paired *t*-test) and Patient L3 39.37 ± 6.81 pmol/min versus 35.57 ± 6.42 pmol/min (*p* < 0.05, paired *t*-test). These rescued ATP levels were similar to those obtained from unaffected control fibroblasts: 42.61 ± 5.94 pmol/min, 51.43 ± 0.63 pmol/min and 44.11 ± 2.19 pmol/min. 

Following the addition of rotenone, AAV-ophNdi1-treated patient fibroblasts demonstrated clear rotenone-resistance (Figure 3H). Unaffected control fibroblasts had rotenone insensitive respiration rates of 14.78 *±* 4.86%, 17.70 *±* 6.22% and 14.17 *±* 8.95%. Patient cells L1, L2 and L3 had rotenone insensitive respiration rates of 9.45 ± 4.12%, 11.65 ± 6.49% and 10.27 ± 2.02%, respectively. Following treatment with AAV-ophNdi1 the rotenone-insensitive respiration rates increased significantly: Patient L1 19.06 ± 10.63% (*p* < 0.05, paired *t*-test), Patient L2 54.99 ± 11.11% (*p* < 0.005, paired *t*-test) and Patient L3 32.74 ± 11.98% (*p* < 0.05, paired *t*-test). These data indicate that AAV-ophNdi1 has indeed provided functional benefit in a patient-derived model of mitochondrial dysfunction.

## 4. Discussion

The retina has a high metabolic requirement making it particularly vulnerable to mitochondrial dysfunction. The critical importance of RGCs to vision, through the processing of visual information, highlights the relevance of preservation of RGCs for many forms of retinal degeneration. Complex I is critical to RGC survival. The large energy demands of RGCs render them particularly reliant on proper functioning of the mitochondria and OXPHOS. Dysfunctional mitochondria not only fail to meet these energy demands but additionally induce oxidative stress through ROS generation. The data presented here clearly demonstrate the potential of ophNdi1 to preserve RGCs by correcting mitochondrial function and decreasing ROS. Given that AAV-ophNdi1 can act to ameliorate mitochondrial dysfunction, the therapy may also have a vital role to play in ocular conditions involving other retinal cell types [75].

In the current study, we have shown AAV-ophNdi1 to be protective in three models of complex I dysfunction. In a rotenone-induced cell model (Figure 1) ophNdi1 performed significantly better than wild type NDI1: expressing more highly, reducing ROS and increasing cellular bioenergetics more significantly. In vivo, AAV-ophNdi1 significantly protected RGCs resulting in improved spatial and functional vision in the rotenone-induced mouse model (Figure 2). Indeed, OKR frequency and PhNR amplitudes in the AAV-ophNdi1-treated model were similar to wild type un-insulted mice. Notably, AAV-ophNdi1 also significantly increased the bioenergetic profile of primary LHON patient-derived fibroblasts with mutations in a subunit of complex I, and significantly increased their resistance to rotenone (Figure 3). These data suggest that AAV-ophNdi1 can ameliorate the electron transport chain dysfunction caused by the underlying complex I mutation and can drive mitochondrial function in the complete absence of complex I. 

NDI1 has previously shown improved OXPHOS with associated therapeutic benefit in rotenone-induced murine models of complex I dysfunction [38,39,76]. In the current study, the significantly increased level of expression afforded by ophNdi1 (Figure 2) offers a means to combat mitochondrial dysfunction with the potential to lower viral dose requirements in future studies in higher species, thereby reducing the chance of AAV immunogenicity [42] and cost of the therapeutic. In principle, the viral capsid and transgene product [77] have the capacity to elicit an immune response. Hence, we employed a dual approach of optimised transgene expression and codon substitution to augment transgene expression and reduce potential immune responses. This may be particularly timely given that adverse events have been reported in the high dose group in a number of studies [78,79,80], including two recent clinical trials [81,82,83]. Furthermore, the significant benefits of ophNdi1 may offer the opportunity to attain functional responses in other studies using NDI1 where optimal expression may have been a limiting factor.

Lower dose requirements may also open up the possibility of utilising AAV-ophNdi1 to treat other mitochondrial/OXPHOS disorders where systemic delivery will be a necessity due to multiple organ involvement, including MELAS, MERF, NARP [84] and MS [85], among others. This would likely require an AAV serotype capable of transducing multiple tissues and able to cross the blood brain barrier (BBB) to the CNS, such as AAV9 [86,87,88], AAV9-2YF [89], AAV-F [90], AAV.PHP.B, AAV.PHP.eB [91,92,93,94] or AAV.CAP-B10 [95], among others. 

Antioxidants and mitochondrial cocktails (e.g., CoQ10, folic acid and ascorbic acid) are widely used in patients with mitochondrial diseases, although robust efficacy has typically not been demonstrated in clinical trials [96]. In the eye, a number of potential treatments to promote normal bioenergetics and preserve RGCs have recently progressed to late stage clinical trials, including Renexus^®^ NT-501, a ciliary neurotrophic factor (CNTF) secreting implant for glaucoma (Phase II, reporting end 2023, clinicaltrials.gov), and Elamipretide, an apoptosis inhibitor that reduces the release of cytochrome C and mROS (Phase II ReSIGHT, Stealth BioTherapeutics; [97,98]), among others. Notably, Raxone^®^, a synthetic CoQ10 analogue was approved by the European Medicines Agency (EMA) to treat LHON in 2015 [99,100], and LUMEVOQ^®^, an allotopic *ND4* gene replacement therapy for the most common LHON mtDNA mutation (m.11778G>A; MT-*ND4*), received a temporary authorisation for use in France in 2021 (EURETINA) (Phase III RESCUE, REVERSE, REFLECT, GenSight Biologics; [28,101,102]). Both therapies showed the most benefit when analysed 2–4 years post-treatment [103,104], highlighting the necessity for robust natural history studies to determine treatment effects in rare diseases. 

We have optimised NDI1 and have shown ophNdi1 to be significantly more potent than wild type NDI1 in a murine and two cell models of complex I deficiency. In each model significant and enhanced benefit was achieved. As a single gene complex I equivalent, ophNdi1 offers a general therapeutic approach for conditions with complex I deficiency regardless of the underlying genetic cause. The proposed approach holds great therapeutic promise for treating conditions where mitochondrial dysfunction plays a significant role.

## 5. Patent 

Farrar, G.J., Millington-Ward, S., Chadderton, N., Carrigan, M.A., Kenna, P.F., Variants of yeast NDI1gene, and uses thereof in the treatment of disease associated with mitochondrial dysfunction. Patent number: 10220102. Filed: 21 December 2012. Date of Patent: 5 March 2019.

## Figures and Tables

**Figure 1 pharmaceutics-15-00322-f001:**
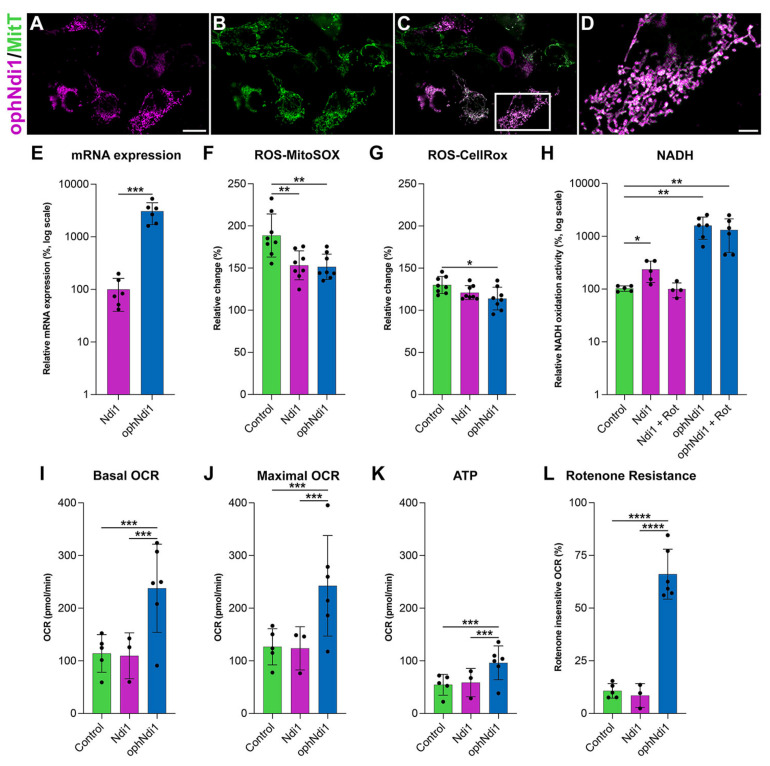
Functional analyses of ophNdi1 in vitro. To establish mitochondrial localisation of ophNdi1, HEK293 cells were co-transfected with 0.5 μg HA-tagged ophNdi1 and analysed 48 h later using Cy3-labelled HA immunocytochemistry (**A**) and MitoTracker™ (**B**); ophNdi1 and MitoTracker™ are co-localised (**C**,**D**). Bioenergetic profiles were evaluated in HEK293 cells transfected with 1 μg pAAV-MCS (control, green bars), 1 μg pAAV-NDI1 (pink bars) or the copy number equivalent of pAAV-ophNdi1 (blue bars) (**E**–**H**); or transduced with AAV-NDI1 (pink bars) or AAV-ophNdi1 (blue bars) at an MOI 3.4 × 10^5^ and compared to untransduced cells (green bars) (**I**,**J**). Bar charts represent: (**E**) Relative mRNA expression levels in HEK293 cells harvested 24 h post-transfection of pAAV-NDI1 or pAAV-ophNdi1. RNA was isolated and mRNA expression analysed by RT-qPCR. Relative expression of NDI1 was assigned a value of 100. (**F**) Cellular (CellRox™) and (**G**) mitochondrial (MitoSox™) ROS levels measured 48 h post-transfection. Cells were insulted with 5 nM rotenone for 24 h prior to analysis or left un-insulted. The percentage increase in ROS between rotenone-insulted and un-insulted cells is shown. (**H**) The rate of NADH oxidation analysed 72 h days post-transfection, normalised to protein. Null transfected control cells, without rotenone addition, were taken to be 100%. (**I**–**L**) Mitochondrial stress test (Seahorse XFe96 Analyser, Agilent) was performed 48 h post-transduction of AAV-NDI1 or AAV-ophNdi1 (*n* = 5). Oligomycin (1.0 µM), FCCP (1.0 µM), rotenone (0.5 µM) and antimycin A (0.5 µM) were injected sequentially and the resultant oxygen consumption rates (OCRs) normalised to protein. Bar charts represent: (**I**) Basal OCR, (**J**) Maximal OCR, (**K**) ATP production, (**L**) Rotenone-insensitive OCR. Error bars represent SD values, * *p* < 0.05, ** *p* < 0.01, *** *p* < 0.001, **** *p* < 0.0001 (**E**: Student’s *t*-test, **F**–**K**: one way ANOVA with Tukey’s multiple comparisons post-hoc test). Scale bar (**A**): 20 µm, (**D**): 5 µm.

**Figure 2 pharmaceutics-15-00322-f002:**
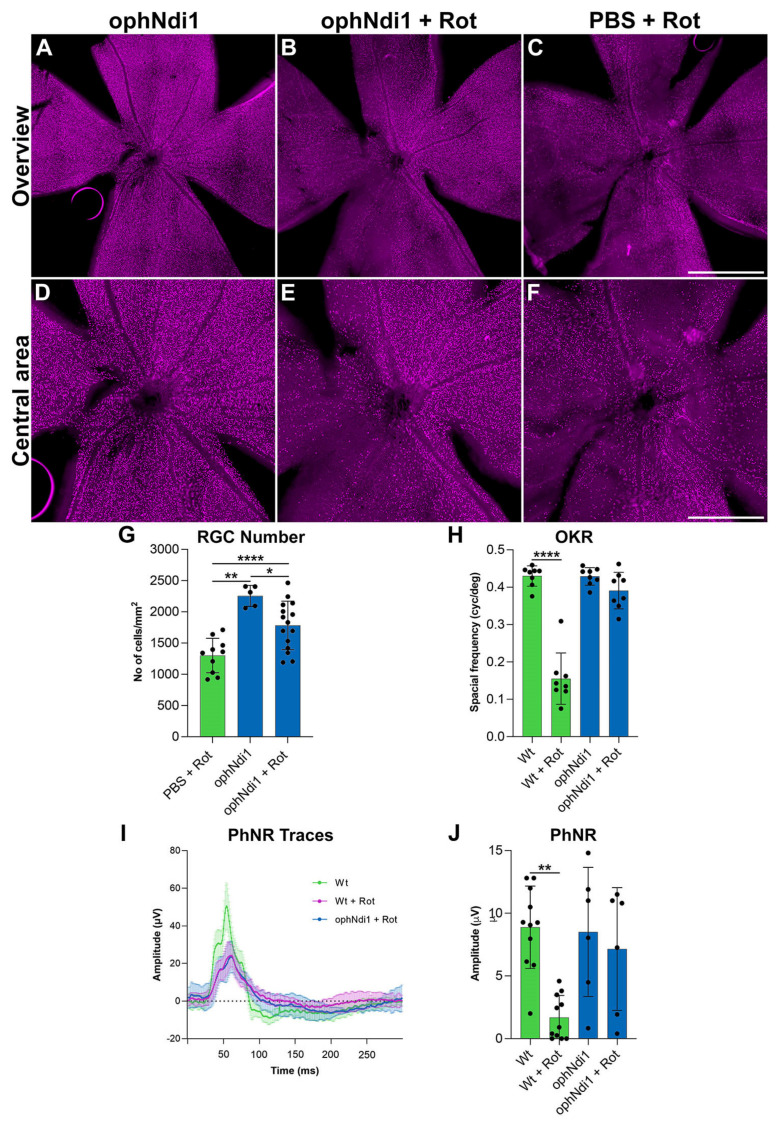
Analysis of AAV-ophNdi1-treated mice following rotenone insult. To establish the functional effects of AAV-ophNdi1 in vivo, adult 129 S2/SvHsd wild type mice were injected intravitreally (IVT) with 3 × 10^9^ vg/eye AAV-ophNdi1 (blue bars) or PBS (green bars), plus 1 × 10^8^ AAV-EGFP, and 1.5 mM rotenone was administered IVT six weeks later, as indicated. Bar charts represent: (**H**) Five weeks post-rotenone treatment optokinetic responses (OKR) were measured (OptoMotry, Cerebral Mechanics [55]). Bar chart represents mean spatial frequency threshold established per eye, *n* = 8–11. (**I**,**J**) Seven weeks post-rotenone treatment the amplitude of the photonegative response (PhNR) was measured (Roland Consult RetiScan). Bar chart represents individual PhNR amplitudes established per eye (*n* = 6–11). (**A**–**G**) Eight weeks post-rotenone treatment eyes were enucleated and fixed in 4% pfa and wholemounts prepared. Wholemount retinas were stained for BRN3A (Cy3 label) immunocytochemistry (**A**–**F**). Retinal ganglion cells (RGCs) were automatically quantified using cellSens software (**G**; Olympus). Scale bar (**C**): 1000 µm, (**F**): 500 µm. Error bars represent SD values; * *p* < 0.05, ** *p* < 0.01, **** *p* < 0.0001 (one way ANOVA with Tukey’s multiple comparisons post-hoc test).

**Figure 3 pharmaceutics-15-00322-f003:**
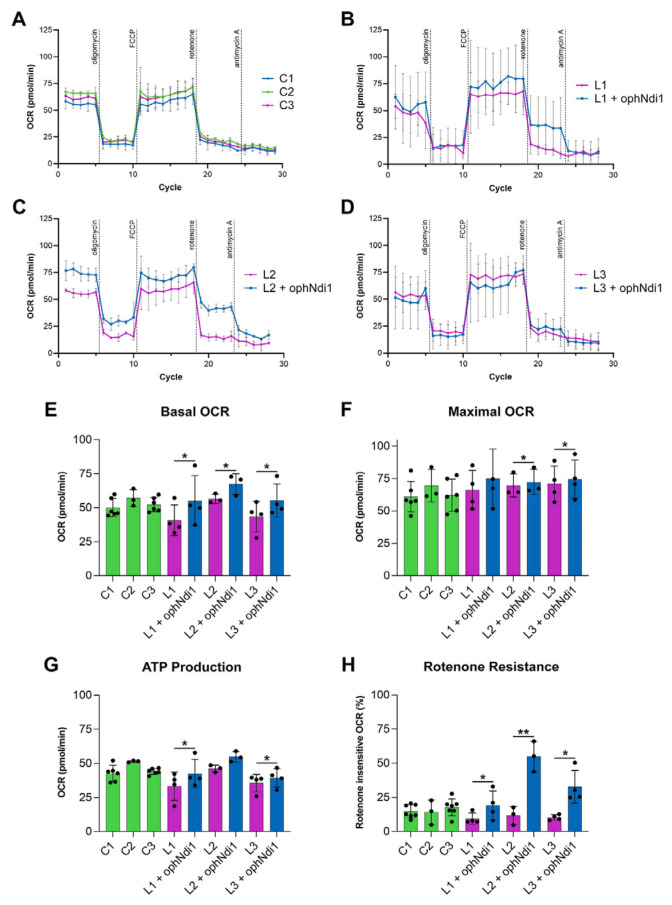
Bioenergetic analysis of AAV-ophNdi1 in LHON patient-derived fibroblasts. Fibroblasts transduced with AAV-ophNdi1 (blue bars) at an MOI of 3.4 × 10^5^ were compared to untransduced patient fibroblasts (pink bars) and untreated unaffected age matched control fibroblasts (green bars), *n* = 5. At 48 h post-transduction a mitochondrial stress test was performed on a Seahorse XFe96 Analyser. Oligomycin (1.0 µM), FCCP (2.25 µM), rotenone (0.5 µM) and antimycin A (0.5 µM) were injected sequentially and the resultant oxygen consumption rates (OCRs) normalised to protein (**A**–**D**). Bar charts represent: (**E**) Basal OCR, (**F**) maximal OCR, (**G**) ATP production and (**H**) rotenone-insensitive respiration. Error bars represent SD values, * *p* < 0.05, ** *p* < 0.01, (Student’s *t*-test).

## Data Availability

All data are available in the main text or the Appendix As.

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
