# Peer review of "Optimisation of AAV-NDI1 Significantly Enhances Its Therapeutic Value for Correcting Retinal Mitochondrial Dysfunction"

_pharmaceutics, 2023, doi:10.3390/pharmaceutics15020322_

Round 1
Reviewer 1 Report
In this paper, the authors optimized the NDI1 and showed ophNdi1 was significantly more potent than wild type NDI1 in a murine and two cell models for complex I deficiency. The manuscript is well organized and well written, and results are interesting. This study may provide potential evidence of AAV-ophNdi1 as a candidate therapy for ocular primary mitochondrial disorders. The paper will be a good contribution to the field and fit well with the scope of the Journal. But there are still some issues need to be addressed.
1. OphNdi1 was initially evaluated in HEK293 cells. It will be better to test it in RGCs as well.
2. The authors predicted that optimized ophNdi1 reduced potential immunogenicity. Is there any data to support the prediction?
3. The font in Figure S2 is too small for reading.
Author Response
Please see the attached file for detailed in line responses to the comments

Reviewer 2 Report
Major:
The authors present an optimized version of their previous Ndi1 vector. Their data demonstrated that the new version apparently is more efficient but in vivo function comparison was not performed.
- The author should mention the sequence of the ophNdi1 vector used.
- Figure 1. Authors should place side by side the ICC picture of NdiI and ophNdil. Similarly, show that the tagged version of the ophNdi1 is located near wildtype Ndi1.
- Authors should explain why they performed some experiments with the pAAV plasmids and AAV vectors. Also, it is essential to clarify why the different time points were used (24, 48 and 72H).
- The colours of the graphic should be changed in order to better discriminate the group at the moment all bars are blue. The individual point should be made visible (use dot plots).
- Please show the complete OCR graphics e.g. as in Fig. 4C from doi.org/10.1016/j.bbrc.2022.10.009 to allow better visualization of the quality of the measurements.
- Please show the levels of both proteins after AAV injection in the eye. Only mRNA was shown. Is the localization/distribution of both proteins the same? Please show ICC of the localization of the proteins in the retina.
- Figure 2. Please rearrange the figure by showing first the control then the Ndi1 and then the ophNdi1. The same goes for WT, WT+rot, ophNdi1, ophNdi1+rot. Once the authors are comparing both vectors, also in the rescue experiments this comparison should be made (RGCs number, OKT and ERG).
- Please show the Phnr traces.
Minor:
- Line 63: Please mention the percentage of the cases.
- Lines 141-143: The sentence does not make sense.
- Line 272: "Cells were insulted with 0nM..." please rephrase the sentence.
Author Response
Please see the attachment for detailed inline response to the comments

Round 2
Reviewer 1 Report
In their revised manuscript, the authors have responded to the comments provided in the original one. They also included more discussions for helping comprehensively explain the results. The paper will be a good contribution to the field.
Reviewer 2 Report
the authors have addressed all my comments